# Trends in Antimicrobial Resistance Patterns in Neisseria Gonorrhoeae in Australia and New Zealand: A Meta-analysis and Systematic Review

**DOI:** 10.3390/antibiotics8040191

**Published:** 2019-10-23

**Authors:** Stephanie Fletcher-Lartey, Mithilesh Dronavalli, Kate Alexander, Sayontonee Ghosh, Leng Boonwaat, Jane Thomas, Amanda Robinson, Zeel Patel, Bradley Forssman, Naru Pal

**Affiliations:** 1Public Health Unit, South Western Sydney Local Health District, Liverpool, NSW 2170, Australia; Mithilesh.Dronavalli@health.nsw.gov.au (M.D.); Kate.Alexander3@health.nsw.gov.au (K.A.); Sayontonee.Ghosh@health.nsw.gov.au (S.G.); Leng.Boonwaat@health.nsw.gov.au (L.B.); Zeel.Patel@health.nsw.gov.au (Z.P.); Naru.Pal@health.nsw.gov.au (N.P.); 2Public Health Unit, Nepean Blue Mountains Local Health District, Penrith, NSW 2750, Australia; Jane.Thomas@health.nsw.gov.au (J.T.); Amanda.Robinson2@health.nsw.gov.au (A.R.); Bradley.Forssman@health.nsw.gov.au (B.F.)

**Keywords:** *Neisseria gonorrhoeae*, sexually transmitted infections, surveillance, antibiotic resistance, antimicrobial testing, Australia, New Zealand

## Abstract

(1) Background: The widespread development of resistance among *Neisseria gonorrhoeae* (NG) clinical isolates has been reported by surveillance systems around the world. This meta-analysis estimated the changes in susceptibility patterns among antibiotics under surveillance in Australia and New Zealand. (2) Methods: Articles published in English from 1980–2018, from Australia or New Zealand, that met the selection criteria were included. The meta-analysis was carried out using the R statistical software. (3) Results: In Australia, there has been decreasing susceptibility of gonococcal isolates to selected antimicrobials over time. Azithromycin (Odds Ratio (OR): 0.73; 95% Confidence Interval (CI) 0.64–0.82) and ceftriaxone (OR: 0.69; 95% CI 0.59–0.80) showed decreasing levels of susceptibility each year. Western Australia (OR: 0.76; 95% CI 0.60–0.96) and Victoria (OR: 0.74; 95% CI 0.60–0.90) also had decreasing levels of susceptibility to ceftriaxone over time compared with other states and territories. (4) Conclusions: The results highlight the need for the development of new approaches for managing cases of gonorrhoea. Improved antimicrobial stewardship, enhanced surveillance and contact tracing are needed to identify and respond to changes in antibiotic resistance in a timely manner. Increasing awareness and public health follow-up of cases can help to interrupt the cycle of infection and limit transmission.

## 1. Introduction

Increasing antimicrobial resistance to *Neisseria gonorrhoeae* has been reported worldwide, including to extended spectrum cephalosporins (ESCs)—the last remaining options for empirical therapy. The World Health Organization (WHO), the US Centres for Disease Control and Prevention (CDC) and the European Centre for Disease Prevention and Control (ECDC) consider the risk of outbreaks due to multidrug-resistant (MDR) and extensively drug-resistant (XDR) gonococci to be a major global public health threat [1,2,3].

The WHO Gonococcal Antimicrobial Surveillance Programme (GASP) data from 2009 to 2014 indicates that resistance is widespread to penicillin, tetracycline and ciprofloxacin. Increasing resistance is being detected for azithromycin and ESCs (ceftriaxone and cefixime), resulting in the WHO recommending dual antimicrobial therapy with ceftriaxone and azithromycin as the optimal treatment for *N. gonorrhoeae*. Although in many countries ceftriaxone is used as monotherapy for treatment of *N. gonorrhoeae*, many developed countries have now implemented a combination therapy with ceftriaxone and azithromycin to combat this problem [4,5,6]. Since 2016, XDR gonococci strains which are resistant to azithromycin and ceftriaxone dual therapy have emerged in some countries [6,7], including many with well-resourced health systems [4,8].

### 1.1. Antimicrobial Resistance(AMR) and the Environment

The emergence of drug-resistant *N gonorrhoeae* is not isolated and falls within an overall global epidemic of antibiotic resistance, which has emerged across many bacterial genera. Current evidence suggests that widespread dependency on antibiotics and complex interactions between human health, animal husbandry and veterinary medicine have contributed to the propagation and spread of resistant organisms [9].

Early reports of the development of antibiotic resistance in hospitals where the majority of drugs were being prescribed was reported among organisms such as sulfonamide-resistant *Streptococcus pyogenes* in the 1930s [10] and penicillin-resistant *Staphylococcus aureus* in the 1940s [11]. Resistance to streptomycin was observed in *Mycobacterium tuberculosis* soon after its discovery [12], and this was followed by the emergence of multiple drug resistance among pathogens such as *Escherichia coli*, *Shigella* and *Salmonella* [13] in the 1950s and 1960s. The increasing prevalence of resistance to multiple antibiotics continued to spread among pathogens such as *Haemophilus influenzae* [14,15,16], and in the 1970s, ampicillin-resistant *Neisseria gonorrhoeae* emerged in developed countries [17,18].

Although microorganisms develop antimicrobial resistance (AMR) as part of their natural evolutionary process [18], this can be accelerated by environmental factors. These include selective pressure due to the widespread frequent use of antimicrobials, through resistance mechanisms transferred to humans from bacteria through food, water or the environment [9,19]. The versatility of *N. gonorrhoeae* to develop resistance mechanisms and to evade host defences has resulted in the global epidemic of resistance among this organism [1,2,3]. The development of antimicrobial resistance and the mechanisms through which they spread has been extensively discussed in existing literature [10,15,16,19]. Going forward, the remainder of this paper will focus on surveillance for antimicrobial resistance in *N. gonorrhoeae*.

### 1.2. Surveillance for Antimicrobial Resistance in Neisseria Gonorrhoeae

The WHO and US National Committee for Clinical Laboratory Standards (NCCLS) recommends that laboratory-based surveillance be carried out to measure resistance to *N. gonorrhoeae*. These recommendations include testing resistance to penicillin, tetracycline and the production of ß-lactamase. Testing for ß-lactamase production allows for the classification of penicillin resistance as a specific beta-lactamase—penicillinase (referred to as penicillinase-producing *N. gonorrhoeae*—PPNG) or chromosomally-mediated (referred to as chromosomally-mediated resistance to penicillin—CMRNG or CMRP) [20].

Changes to the minimum inhibitory concentration (MIC) value (mg/L) denote that multiple and different chromosomal changes are present in an organism. Chromosomal resistance is classified as susceptible (MIC 0.03 mg/L), reduced susceptibility (MIC 0.06–0.5 mg/L) or relatively resistant (MIC 1 mg/L). PPNG are a separate resistance category, and therefore, are not measured in MIC values. Gonococcal infections that are susceptible or reduced susceptibility usually respond to penicillin-based treatments. PPNG and relatively resistant strains usually fail to respond to penicillin. The WHO recommends changing antibiotic regimens when the rate of resistance to an antibiotic is five percent or more [21]. 

The gold standard for determining MIC values is the agar dilution method, as accepted by the WHO. In Australia, the Calibrated Dichotomous Sensitivity (CDS) is used which utilises the agar dilution method [22]. 

### 1.3. The World Health Organization Global Gonococcal Antimicrobial Surveillance Programme (WHO GASP)

In 1990, the WHO Global Gonococcal Antimicrobial Surveillance Programme (WHO GASP) was established to monitor gonococcal antimicrobial resistance (AMR) worldwide. The WHO GASP has continued to develop public health strategies and clinical guidelines with increasing prominence since 2009. The WHO GASP develops these strategies through surveillance systems with external and internal quality control, including utilising regional technical support and training to strengthen laboratory capacities for accuracy and compatibility [23]. The WHO GASP works in coordination with other GASPs including Europe [24,25], USA [26], Canada [27], Australia [28,29] and the UK [29].

The WHO Coordinating Centre in Sydney has coordinated the surveillance of gonococcal resistance in Australia since 1992, initially for ciprofloxacin and penicillin and subsequently for ceftriaxone and azithromycin. Due to different patterns of resistance in remote versus non-remote populations in Australia, data is de-aggregated and separately reported. Survey data is collated at the state and territory or country level for all other sites [30].

### 1.4. Australian Gonococcal Surveillance Programme (AGSP)

The Australian Gonococcal Surveillance Programme (AGSP) monitors *Neisseria gonorrhoeae* AMR throughout all Australian states and territories, and informs clinical guidelines for the treatment of gonococcal infection [31]. Antibiotics surveyed include penicillin, ceftriaxone, ciprofloxacin, spectinomycin, tetracycline, and azithromycin. Since 1981, AGSP has utilised standardised lab methods and reporting [22,32] to facilitate accurate comparisons over time and between states and territories in Australia [33]. A quality assurance programme is in place with reference laboratories in several states submitting data quarterly to a coordinating laboratory that collates the results and conducts quality assurance [33]. Reporting includes site of isolation and gender for gonococcal cases, and the geographic source of acquisition for resistant strains where possible [34].

This systematic review, meta-analysis and meta-regression describes the Australian gonococcal surveillance system and the changes in susceptibility patterns for the main classes of antibiotics used in the empirical treatment for gonorrhoea over time and location. The results highlight the need for the development of new approaches to the management of gonorrhoea. Improved antimicrobial stewardship, enhanced surveillance and contact tracing are required to identify and respond to changes in antibiotic resistance in a timely manner. Increasing awareness and public health follow-up of cases can help to interrupt the cycle of infection and limit transmission.

## 2. Results

A total of 2848 potential abstracts were identified, and 212 articles of relevance were screened. After assessing 42 articles for eligibility, 29 articles [28,31,34,35,36,37,38,39,40,41,42,43,44,45,46,47,48,49,50,51,52,53,54,55,56,57,58,59,60,61,62,63] that met the criteria were included in the meta-analysis. Thirteen studies [45,47,64,65,66,67,68] were excluded, including four studies that presented duplicate data and five molecular studies that examined less than 100 isolates or did not provide sufficient data on gonococcal susceptibility in keeping with the inclusion criteria (Figure 1).

### 2.1. Study Characteristics

The studies included were published between 1980 and 2017. The main characteristics of the 29 included studies are presented in Table 1. Studies conducted prior to 1996 included two that presented data from the AGSP that summarised antimicrobial sensitivity data for the period 1981–2017 for Australia. Two studies included data from only Sydney, Australia. Annual reports of the AGSP published in the Communicable Diseases Intelligence Journal covered the period 1996–2017. Data from three studies carried out in New Zealand were also reviewed as a means of comparison. 

### 2.2. N. gonorrhoeae Resistance to Antimicrobials

The summary of the number of isolates tested, number of isolates susceptible, reduced susceptibility and resistance are presented in Appendix A. The summary data on antimicrobial resistance patterns and the odds ratio for changes in rates in Australia, Australian states and territories and New Zealand over time, are presented in Appendix A. Figure 2 outlines the meta-regression plots showing changes in the proportion of isolates susceptible to each antibiotic over time in Australia, Australian states and territories and New Zealand. Meta-regression plots showing proportions of isolates with reduced susceptibility to selected antibiotics (Figure 3) and high levels of resistance (Figure 4) in Australia, by Australian states and territories, and New Zealand are also presented. The trends in susceptibility rates to various antimicrobials are summarised below. 

#### 2.2.1. Susceptibility to Penicillin

Data for penicillin was available from 19 studies covering the periods 1981–1985, 1987–1993, 1997, 1999, 2000–2001, 2003, 2005, 2009–2017. In Australia, the proportion of isolates susceptible to penicillin was on average 43.1% (95% CI 27.8–59.9%). The analysis revealed fluctuations in the susceptibility rates (OR: 1.03; 95% CI 0.98–1.09), with an overall increase over time (Figure 2). In comparison, data from New Zealand covering the periods 1988, 2003 and 2015 revealed significant decreases in penicillin susceptibility over time (OR: 0.74; 95% CI 0.72–0.78).

There was significant variation in the proportion of isolates demonstrating significant levels of resistance to penicillin over time. Significant changes in the levels of resistance over the period were observed in Queensland (OR: 1.03; 95% CI 1.01–1.05) and the Northern Territory (OR: 1.03; 95% CI 1.00–1.06).

A larger proportion of resistant isolates demonstrated chromosomally-mediated resistance (CMRP) to penicillin compared with penicillinase producing *N. gonorrhoeae* (PPNG). Meta-regression plots showing the weighted average proportion of isolates that were PPNG (Appendix A) and that demonstrated CMRP (Appendix A) over time in Australia, Australian states and territories and New Zealand are included in the Appendix A. The proportion of PPNG fluctuated between 1.0% (0.0%–1%) and 22.0% (16.0%–29.0%) over the period 1993–2017, with an average proportion of 14% (6.0%–29.0%) PPNG over the period 1993–2017. The average proportion of CMRP was on average 24.0% (10.0%–49.0%) and fluctuated between the range 9.0% (0.9%–10%) and 25.0% (24.0%–26.0%) over the period 1996–2017. One New Zealand study [46] found up to 59% of isolates in 2015 were CMRP.

#### 2.2.2. Susceptibility to Ceftriaxone

Ceftriaxone data was reported by 18 studies covering the periods 1993, 1996–97, 1999, 2000, and 2010–17. In Australia, the proportion of isolates susceptible to ceftriaxone was on average 99.7% (95% CI 98.3%–99.9%). The rate of susceptibility decreased over the period (OR: 0.69; 95% CI 0.59–0.80) (Figure 2). There were also significant increases in the proportion of isolates demonstrating reduced susceptibility to ceftriaxone across Australia (OR: 1.52; 95% CI 1.24–1.86) (Figure 3). Over the period, the states having the most significant increases in reduced susceptibility were Western Australia (OR: 1.40; 95% CI 1.04–1.88) followed by Victoria (OR: 1.36; 95% CI 1.11–1.67), New South Wales (NSW) (OR: 1.36; 95% CI 1.06–1.74) and Queensland (OR: 1.25; 95% CI 1.03–1.53).

#### 2.2.3. Susceptibility to Ciprofloxacin

Ciprofloxacin data from 19 studies covered the periods 1986, 1991–93, 1997, 1999, 2000–01, 2005–06, and 2010–17. In Australia, the proportion of isolates susceptible to ciprofloxacin decreased over time (OR: 0.90; 95% CI 0.87–0.92), with an average susceptibility of 86.4% (95% CI 79.6%–91.3%). The trend of decreasing proportions susceptible over time was evident across all Australian sand territories, with the greatest decreases over the period in the Northern Territory (OR: 0.88; 95% CI 0.84–0.93) and the lowest in New South Wales (OR: 0.95; 95% CI 0.91–0.99) (Figure 2). Data from New Zealand covering the periods 1988, 2003 and 2015 revealed an even greater decrease in susceptibility to ciprofloxacin over the period (OR: 0.80; 95% CI 0.68–0.93).

There was significant variation in the proportion of isolates demonstrating resistance to ciprofloxacin across Australia (OR: 1.23; 95% CI 1.16–1.31) over time (Figure 4). Significant changes in the levels of resistance over the period were observed in all states and territories, with the greatest increases over time observed in South Australia (OR: 1.14; 95% CI 1.08–1.19), the Northern Territory (OR: 1.13; 95% CI 1.08–1.19) and Western Australia (OR: 1.13; 95% CI 1.07–1.19). Higher rates of resistance (OR: 1.17; 95% CI 1.13–1.21) were observed in New Zealand over time.

#### 2.2.4. Susceptibility to Tetracycline

Thirteen studies reported on susceptibility to tetracycline covering 1993, 1996, 1999, 2000, 2005–06, and 2010–2017. In Australia, there was evidence of decreasing susceptibility to tetracycline (OR: 0.95; 95% CI 0.93–0.98) over time, with, on average, 88.7% (95% CI 85.7%–91.2%) of isolates remaining susceptible over the period. The decreasing trend was evident at the state level, with the highest levels of decreasing susceptibility observed in the Northern Territory (OR: 0.76; 95% CI 0.69–0.83). Data available from two studies conducted in New Zealand covering 1998 and 2015 found a similar trend of significant reduction in susceptibility rates (OR: 0.87; 95% CI 0.85–0.88) over time.

The rate of resistance to tetracycline across Australia (OR: 1.05; 95% CI 1.02–1.08) was increasing with time (Figure 4). Significant changes in the levels of resistance were observed in all states and territories (OR range from 1.06 (95% CI 1.03–1.11) in Western Australia to 1.31 (95% CI 1.20–1.45) in the Northern Territory) except for Victoria. The rate of resistance (OR: 1.30; 95% CI 1.00–1.68) observed in New Zealand over time was even greater.

#### 2.2.5. Susceptibility to Azithromycin

Data for azithromycin was available for 1997 and 2012–17, with evidence of complete susceptibility in 1997 and declining susceptibility thereafter. In Australia, the proportion of isolates susceptible to azithromycin remained relatively high (OR: 97.3%; 95% CI 95.6%–98.3%) and the analysis revealed decreasing rates of susceptibility (OR: 0.73; 95% CI 0.64–0.82) over time. The highest levels of decreasing susceptibility to azithromycin over the period was observed in South Australia (OR: 0.50; 95% CI 0.33–0.75), followed by New South Wales (OR: 0.57; 95% CI 0.50–0.64) and Western Australia (OR: 0.70; 95% CI 0.59–0.82).

#### 2.2.6. Susceptibility to spectinomycin

Over the period, all studies reported that all isolates tested were susceptible to spectinomycin. 

## 3. Discussion

The Australian Gonococcal Surveillance Programme (AGSP) has monitored the incidence of gonorrhoea and antibiotic sensitivity of gonococci since 1981. The WHO recommends that epidemiological surveillance of the distribution and extent of antimicrobial resistance is necessary to guide treatment regimens for gonorrhoea. The recommendation indicates that surveillance data can be used as a surrogate indicator of the effectiveness of sexually-transmitted infection (STI) risk-reduction activities and interventions and that an antibiotic should not be used when 5% or more of isolates are resistant [21]. This meta-analysis and meta-regression examined susceptibility trends in antimicrobials against *N. gonorrhoeae* in Australia and New Zealand. This study identified significant reductions in the proportions of isolates demonstrating sensitivity to all classes of antibiotics over time, except for spectinomycin. Significant increases in the proportion of resistant isolates were also observed for most antibiotics, with variations in trends across Australian states and territories, with some similarities to trends in New Zealand. These findings support continued surveillance for AMR trends in *N. gonorrhoeae* and reinforce the need for judicious use of antibiotics and the promotion of STI prevention strategies [69]. 

The study identified some fluctuations but demonstrated an overall increase in the proportion of isolates susceptible to penicillin over time, which is inconsistent with reports from other jurisdictions [28,31]. The increase in susceptibility observed in Australia over time is likely due to the fact that penicillin is no longer routinely used to treat gonococcal infections in Australia, except in Western Australia and the Northern Territory where resistance rates have remained low. 

The AGSP Annual Report 2017 [28] reported isolation of resistant strains from <2% in remote settings in the Northern Territory, where populations are predominantly Indigenous Australians and spread is predominantly through heterosexual transmission. Despite high gonorrhoea notification rates and many years of empirical oral therapy in remote regions of Western Australia, penicillin susceptibility rates remain high [70]. The reduction in penicillin resistance may be associated with a reduction in the proportion of isolates with CMRP, as described by Lahra et al. [31] in 2014. The low rate of penicillin resistance observed in remote regions of Western Australia may also be attributed to the continued efficacy of combination dual therapy and their effectiveness against the emergence of multi-resistant *N. gonorrhoeae* strains [70]. The introduction of a nucleic acid amplification testing (NAAT) assay capable of detecting penicillin resistance [70] in the Northern Territory, has provided enhanced surveillance data since 2015 [28] (more details about NAAT below).

A significant reduction was also noted in the proportion of isolates susceptible to ciprofloxacin and tetracycline observed in Australia. The trend was likely influenced by significantly high rates of resistance in ciprofloxacin prior to 2009. Reports indicate, however, that the proportion of isolates demonstrating resistance has decreased from 54% in 2008 to 28% in 2017 [28]. If this pattern continues, the increased susceptibility observed with penicillin may result.

A significant reduction in susceptibility rates was observed for tetracycline in both Australia and New Zealand. High-level tetracycline-resistant *N. gonorrhoeae* (TRNG) (MIC value ≥ 16 mg/L) is used as an epidemiological marker, even though tetracycline is not a recommended treatment for gonorrhoea. Monitoring the movement or importation of TRNG isolates within a geographic region is beneficial to inform public health actions [71]. In contrast to penicillin, significant proportions of isolates resistant to tetracycline have been reported from rural and remote areas in the Northern Territory and Western Australia [28].

All isolates remained susceptible to spectinomycin. This is in keeping with trends across the Asia Pacific region where very low rates of resistance (0.3%) to spectinomycin have been observed [72].

The increase in the number of isolates with reduced susceptibility to ceftriaxone is a cause for concern. In many countries, ceftriaxone is the only remaining drug used to treat for gonorrhoea. An increase in the proportion of isolates with reduced susceptibility to ceftriaxone was observed in Australia from 2006 to 2013, where proportions increased from <1–9%. However, a reversal in this trend to about 1% was observed in 2017 [28]. Interestingly, *N. gonorrhoeae* isolates with high-level resistance to ceftriaxone (MIC value of 0.5 mg/L) were reported in Australia for the first time in 2017 [41]. The two isolates were also resistant to penicillin (PPNG; MIC ≥ 32 mg/L) and ciprofloxacin (MIC > 32 mg/L) but susceptible to spectinomycin (MIC 8 mg/L) and azithromycin (MIC 0.25 mg/L) [28]. Close monitoring of the AMR patterns in ceftriaxone is therefore critical, since in vitro resistance and/or treatment failures to ceftriaxone, including failures to treat pharyngeal gonorrhoea, have been reported [73]. This has led to high-income regions or countries implementing the recommended dual antimicrobial therapy with ceftriaxone 250–1000 mg plus azithromycin 1–2 g [73,74]. The relatively low rates of antimicrobial resistance observed to ceftriaxone in Australia are likely indicative of the benefits of treating gonorrhoeae with a combination therapy [1].

This study found that a significantly higher proportion of isolates remained susceptible to azithromycin in Australia and New Zealand. There was an eight-fold increase in the proportion of isolates exhibiting resistance (MICs ≥ 1.0 mg/L) observed over the period 2012–2017 [28] and reports of high-level resistance and treatment failures to azithromycin (MIC value ≥ 256 mg/L) [28,31,41,42,44]. The emergence of high-level resistance in azithromycin, particularly in isolates showing resistance to ceftriaxone, would be a major public health issue. While there were only few studies found that reported on azithromycin susceptibility, the results provide evidence for strengthened monitoring of the dual therapy regime in Australia. It suggests the need for ongoing public health vigilance as there is evidence of increasing MICs to azithromycin resistance globally [30,73], and failure of dual therapy has been reported [7].

The number of *N. gonorrhoeae* strains included in antimicrobial resistance testing and captured by surveillance has been impacted by the use of NAAT for diagnosis of gonorrhoea in urban and remote settings [4]. While NAAT assays to detect penicillin resistance have been developed for use in remote regions in Australia, NAAT cannot do wide-scale antimicrobial susceptibility testing [70]. However, in remote settings where delays can reduce the sensitivity and reliability of cultures, NAAT has an advantage. Molecular approaches to AMR testing can also provide important information that helps to establish epidemiological links between cases. However, acknowledging the current limitations of NAAT, they should be used to compliment culture methods to ensure that surveillance systems are able to detect new and emerging resistant strains. 

While the data was not presented in this report, Australian surveillance data indicate that a significant proportion of isolates for which travel history data was available, indicate that infection was either acquired overseas, diagnosed in international travellers or induced by local treatment protocols [28,47,68,75]. It is important to also note, that over the more than two and a half decades covered by data in this study, there have been changes to the recommended prescription protocols, particularly in metropolitan areas, in response to resistance patterns and in keeping with international guidelines [45,76]. The continued efficacy of combination dual therapy and their effectiveness in delaying the emergence of multi-resistant *N. gonorrhoeae* has been described [45,70]. It is therefore likely that an important contributing factor to the increase in the proportion of resistant strains over time is the increased frequency of and continued introductions of resistant strains into the local population rather than local treatment protocols that are driving resistance levels. It is reasonable to assume that some of the noise in the data may reflect the contribution of overseas-acquired infections to the local picture.

The implications of this study highlight the need for alternative methods to prevent and control *N. gonorrhoeae* infections. One recommendation is to devote more effort towards effective preventive measures. To this end, barriers to the uptake of prevention actions need to be anticipated and strategies developed to overcome them. These barriers include a low level of awareness among groups who are at risk of sexually-transmitted infections, social stigma and the need for effective training for health care workers, for example in behavioural counselling. Peterman et al. [46] suggest that one measure to reduce transmission is to advocate for giving patients medication for both themselves and their partners to reduce re-infection rates. This would require legislative changes that allow doctors to do this but may not be practical in cases where injectable antibiotic treatment is in use. They also recommended including screening for STIs when testing for other non-transmissible conditions, but this may require changes to policy and current practices which may not be readily implemented at sexual health clinics. Increasing the availability of STI testing options may also reduce transmission rates. Rapid testing has the potential for reducing numbers lost to follow up as test results are available almost immediately, treatment and partner notification can occur during the same visit and treatment compliance is encouraged [77]. Increasing venues where testing is available should also be considered, particularly in areas frequented by groups who are at risk of sexually-transmitted infections. 

## 4. Materials and Methods 

This meta-analysis was conducted in accordance with the checklist of the Meta-Analysis of Observational Studies in Epidemiology Guideline [78].

### 4.1. Literature Search

Google Scholar was searched to identify relevant studies published in English, from 1980–2019, based on the search term “*Neisseria gonorrhoeae*” OR “gonorrhoea” OR “Gonococcus” AND “antimicrobial” OR “antibiotic” AND “resistance” OR “susceptible” or “susceptibility” AND “Australia”. Relevant references from each study were reviewed to identify additional studies.

#### 4.1.1. Inclusion Criteria

The included studies met the following criteria:A study or report published in English from January 1980 to February 2019 presenting results for 1981–2017;Included data from Australia or any state or territory in Australia or New Zealand;Described the method for determining the antimicrobial susceptibility of isolates or the MIC values for susceptibility, reduced susceptibility and resistant isolates (following the criteria by WHO Western Pacific Region (WPR) Resistance Surveillance Programme guidelines [79] or Clinical and Laboratory Standards Institute (CLSI) standards);Specified the total number of tested *N. gonorrhoeae* isolates;Reported the antimicrobial susceptibility rate or proportion in *N. gonorrhoeae* isolates or implied it by indicating their minimum inhibitory concentrations (MICs) value or the number or proportion of non-susceptible NG isolates;Tested 100 or more *N. gonorrhoeae* isolates, andIsolated *N. gonorrhoeae* strains from human clinical samples.

#### 4.1.2. Exclusion Criteria

Studies were excluded if they were reviews, single case reports, or where results were not presented in a manner to facilitate extraction of data for meta-analysis calculations.

### 4.2. Quality Assessment

The Joanna Briggs Institute Critical Appraisal Checklist for Analytical Cross-Sectional Studies [80] was used to conduct the initial appraisal of the methodological quality of studies. The checklist uses eight criteria to assess the methodological quality of each study and to determine the extent to which each study has addressed the possibility of bias in its design, conduct and analysis. Once the study was considered to be methodologically sound, they were then assessed against the recommended approach of the WHO [20]. Specifically, we assessed the following areas: Whether the study specified the location where *N. gonorrhoeae* isolates were collected;Whether the study specified the collection period of the isolates; Whether the study described the method of identifying *N. gonorrhoeae* isolates; Whether the study described the population from which *N. gonorrhoeae* isolates were obtained;Whether the study included at least 100 tested *N. gonorrhoeae* isolates;Whether the study utilised control strains recommended by WHO in determining MICs or implied it;Whether the study described the method for determining the antimicrobial susceptibility of isolates or the MICs values for susceptibility, reduced susceptibility and resistant isolates.

The quality assessment scores of the included studies were shown in Table 1. Each of the seven criteria were rated one point if a study satisfied it. Two independent reviewers assessed the quality of the included studies. We considered studies that scored five or higher as “high quality”, three or four as “moderate quality”, and two or lower as “low quality”.

### 4.3. Data Extraction

Reports were initially screened for relevance, and all potentially relevant reports were analysed to determine if they satisfied the inclusion criteria. The following data was extracted from each study: name of first author, publication year, study country or region, isolate collection period/year, number of tested isolates, and susceptibility or resistance rate for each antibiotic, and study population (where available). Where percentages were reported, the estimated number of isolates was calculated using the denominator data or proportions reported in the study. Data from the included studies were independently extracted by six authors (SF-L, SG, KA, JT, AR and ZP) and any discrepancies were resolved through consultation with MD and NP. 

### 4.4. Data Analysis

Individual study prevalence statistics were calculated with the number of total isolates as the denominator and each test type as the numerator. Note that the sum of the three isolate test type equals the number of total isolates.

Meta-analysis was carried out using the meta-prop [81] command in R statistical software on all prevalence statistics from studies in the systematic review period by antibiotic (azithromycin, ceftriaxone, ciprofloxacin, penicillin and tetracycline), region (New South Wales, Queensland, Victoria, Australian Capital Territory, Tasmania, South Australia, Northern Territory, Western Australia and New Zealand) and susceptibility classification (susceptible, medium resistance and high resistance).

The meta-analysis results consist of a prevalence statistic with 95% confidence intervals calculated from the weighted average of prevalence statistics for all the studies in the specified sub-group by antibiotic, region and test type. From this meta-analysis of proportions, a summative forest plot was carried out using the forest command [82]. The forest plots from the meta-analysis are available in the Appendix A.

Meta-regression models were used to check for changing antibiotic resistance over time. Models were constructed by antibiotic, region and test type. Results reported from the meta-regression models consist of an odds ratio for the time variable by calendar year. Meta-regression was carried out using the meta-reg command [82]. A statistically significant odds ratio for the time variable indicates that antibiotic resistance was changing over time. 

The meta-regression odds ratio and 95% CI for the time variable and the meta-analysis prevalence statistics with 95% CI are displayed graphically in a non-summative forest plot using the forest plot package [83]. Non-summative forest plots were constructed for susceptibility, medium resistance and high resistance. Antibiotic resistance changing over time indicates that the meta-analysis weighted average of the prevalence of resistance is biased as this prevalence is only cross-sectional and does not account for change over time. 

All statistical analyses was carried out using the statistical package R 3.6.0 (R Foundation for Statistical Computing: Vienna, Austria) [84].

## 5. Conclusions

This study showed that over time, there were changes in antimicrobial susceptibility levels for several antibiotics used in Australia for the management of *N. gonorrhoeae* or used as epidemiological markers. The continued increase in resistance to tetracycline and ciprofloxacin, despite these drugs not being used for routine treatment of gonococcal infections, indicates the ongoing selective pressure produced by the use of these antimicrobials to treat other infections. Improved data collection in place of acquisition is needed to evaluate the impact of overseas-acquired infection on antimicrobial susceptibility trends and to inform public health policies. There is a need to strengthen the monitoring of the dual therapy regime in Australia in light of an increasing trend in reduced susceptibility to ceftriaxone and the emergence of highly resistant strains to azithromycin. There is an urgent need for new approaches, including collaborative efforts to determine how culture-based tests and NAAT may be combined or complemented to strengthen AMR surveillance. These findings also support the need for systematic monitoring of gonococcal treatment failures by developing a standard case definition of treatment failure and enhancement of infectious disease surveillance systems that are guided by protocols for verification, reporting, and management of gonococcal treatment failure in Australia.

## Figures and Tables

**Figure 1 antibiotics-08-00191-f001:**
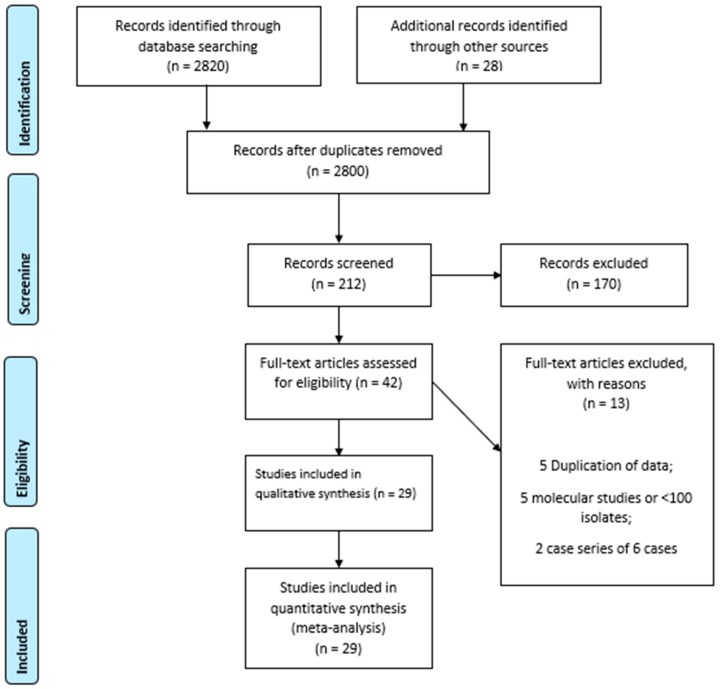
Preferred Reporting Items for Systematic Reviews and Meta-Analyses (PRISMA) 2009 Flow Diagram.

**Figure 2 antibiotics-08-00191-f002:**
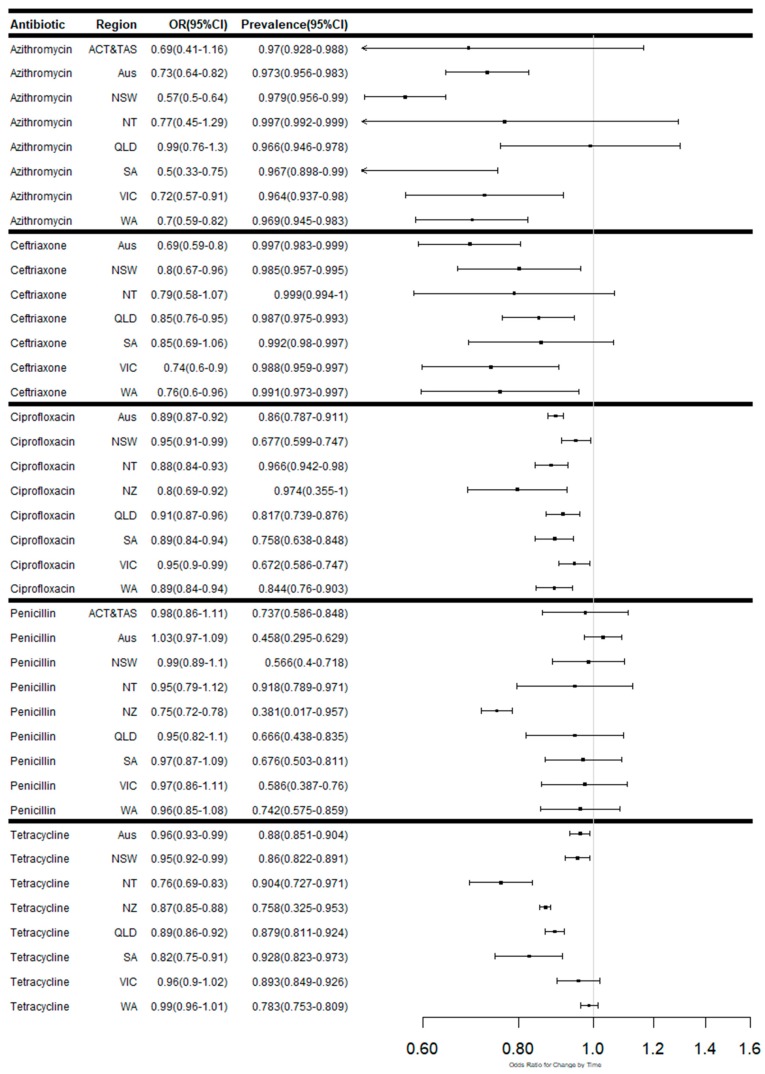
Meta-regression plots showing the odds ratios for changes in proportion and weighted average proportions of isolates susceptible to each antibiotic over time in Australia, Australian states and territories and New Zealand.

**Figure 3 antibiotics-08-00191-f003:**
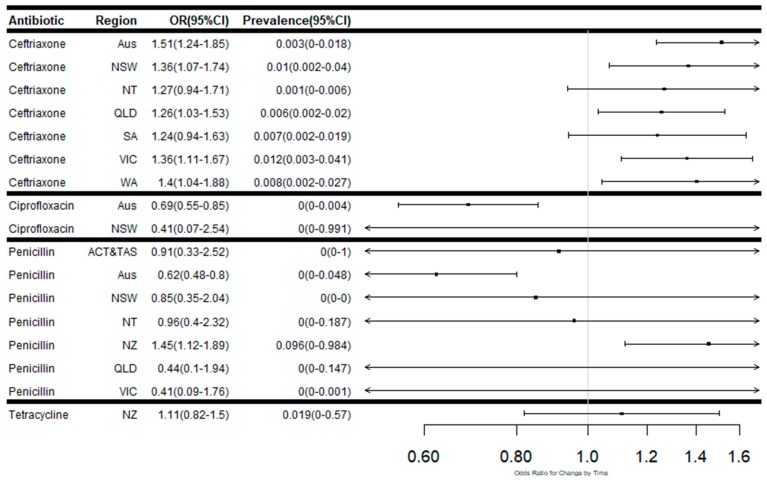
Meta-regression plots showing the odds ratios for changes in proportion and weighted average proportions of isolates with decreased susceptibility to each antibiotic over time in Australia, Australian states and territories and New Zealand.

**Figure 4 antibiotics-08-00191-f004:**
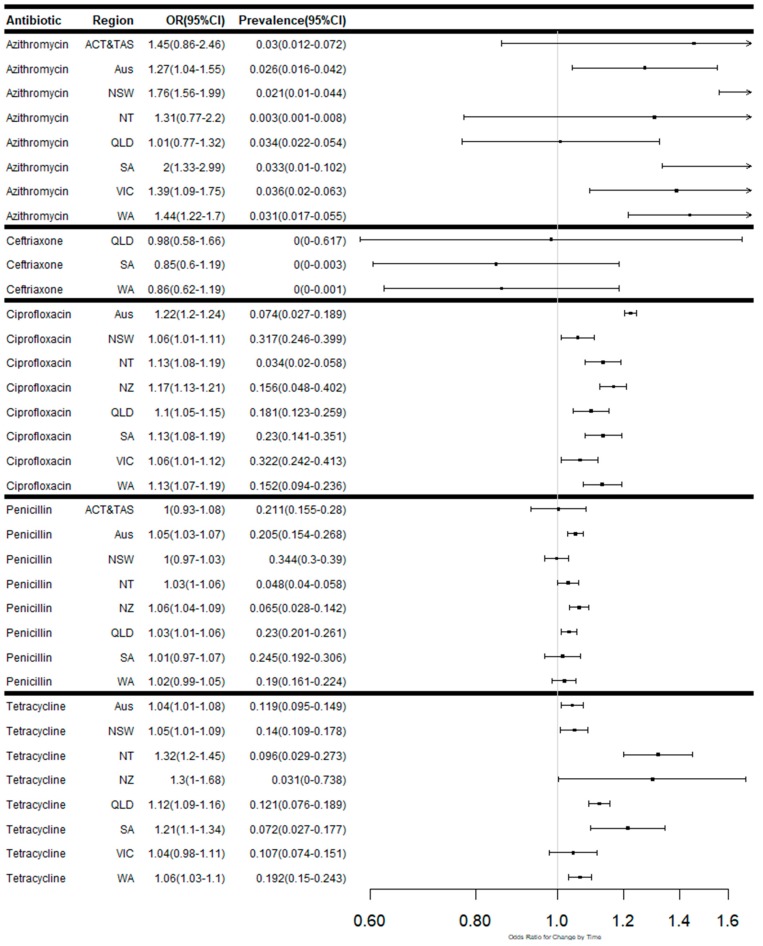
Meta-regression plots showing the odds ratios for changes in proportion and weighted average proportions of isolates resistant to each antibiotic over time in Australia, Australian states and territories and New Zealand.

**Table 1 antibiotics-08-00191-t001:** Summary of the main characteristics of included studies and their respective quality assessment scores.

Author Name (Ref #)	Location	Period of Isolate Collection	Source of Isolates	Subject Sex Identified	Isolation Detection Method	MIC Breakpoints	NG Isolates > = 100	Reference/Control Strains Utilised	Total Number of Samples	Quality Score
Australian Gonococcal Surveillance Programme, 1988 [35]	Australia	1981–1986	Public sector Sexually Transmitted Disease (STD) clinics.	Male and female	Standardised agar plate dilution technique.♦	Standard MIC values for included antimicrobial agents♣	Yes	Australian Gonococcal Surveillance Programme (AGSP) own quality assurance programme used prior to 1993	25,000	6
Tapsall, 1993 [51]	Australia	1981–1991	Mixed population	Male and female	Standardised agar plate dilution technique of AGSP.♦	Standard MIC values for included antimicrobial agents♣	Yes	World Health Organisation (WHO) quality controlled method since 1993.	32,000	7
Tapsall, 1995 [49]	Sydney, Australia	1993	Mixed population	Male and female	Standardised agar plate dilution technique.♦	Standard MIC values for included antimicrobial agents♣	Yes	WHO quality controlled method since 1993.	173	6
Tapsall, 1996 [50]	Sydney, Australia	1991–1995 (+1984–1990)	Mixed population	Male and female	Standardised agar plate dilution technique.♦	Standard MIC values for included antimicrobial agents♣	Yes	WHO quality controlled method since 1993.	2670	6
Tapsall, 1998 [48]	Sydney, Australia	1995–1997	Public and private sector sources	Male and female	Standardised agar plate dilution technique.♦	Standard MIC values for included antimicrobial agents♣	Yes	WHO quality controlled method since 1993.	2236	6
Tapsall, 1997 [34]	Australia	1996	Public and private sector sources	Male and female	Standardised agar plate dilution technique.♦	Standard MIC values for included antimicrobial agents♣	Yes	WHO quality controlled method since 1993.	2753	7
Tapsall, 2000 [52]	Australia	1999	Public and private sector sources	Male and female	Standardised agar plate dilution technique.♦	Standard MIC values for included antimicrobial agents♣	Yes	WHO quality controlled method since 1993.	3658	7
Tapsall, 2001 [53]	Australia	2000	Public and private sector sources	Male and female	Standardised agar plate dilution technique.♦	Standard MIC values for included antimicrobial agents♣	Yes	WHO quality controlled method since 1993.	3468	7
Tapsall, 2002 [54]	Australia	2001	Public and private sector sources	Male and female	Standardised agar plate dilution technique.♦	Standard MIC values for included antimicrobial agents♣	Yes	WHO quality controlled method since 1993.	3725	7
Tapsall, 2003 [55]	Australia	2002	Public and private sector sources	Male and female	Standardised agar plate dilution technique.♦	Standard MIC values for included antimicrobial agents♣	Yes	WHO quality controlled method since 1993.	3861	7
Tapsall, 2004 [56]	Australia	2003	Public and private sector sources	Male and female	Standardised agar plate dilution technique.♦	Standard MIC values for included antimicrobial agents♣	Yes	WHO quality controlled method since 1993.	3677	7
Tapsall, 2005 [57]	Australia	2004	Public and private sector sources	Male and female	Standardised agar plate dilution technique.♦	Standard MIC values for included antimicrobial agents♣	Yes	WHO quality controlled method since 1993.	3664	7
Tapsall, 2006 [58]	Australia	2005	Public and private sector sources	Male and female	Standardised agar plate dilution technique.♦	Standard MIC values for included antimicrobial agents♣	Yes	WHO quality controlled method since 1993.	3886	7
Tapsall, 2007 [59]	Australia	2006	Public and private sector sources	Male and female	Standardised agar plate dilution technique.♦	Standard MIC values for included antimicrobial agents♣	Yes	WHO quality controlled method since 1993.	3850	7
Tapsall, 2008 [60]	Australia	2007	Public and private sector sources	Male and female	Standardised agar plate dilution technique.♦	Standard MIC values for included antimicrobial agents♣	Yes	WHO quality controlled method since 1993.	3142	7
Tapsall, 2009 [61]	Australia	2008	Public and private sector sources	Male and female	Standardised agar plate dilution technique.♦	Standard MIC values for included antimicrobial agents♣	Yes	WHO quality controlled method since 1993.	3110	7
Tapsall, 2010 [62]	Australia	2009	Public and private sector sources	Male and female	Standardised agar plate dilution technique.♦	Standard MIC values for included antimicrobial agents♣	Yes	WHO quality controlled method since 1993.	3157	7
Lahra, 2011 [39]	Australia	2010	Public and private sector sources	Male and female	Standardised agar plate dilution technique.♦	Standard MIC values for included antimicrobial agents♣	Yes	WHO quality controlled method since 1993.	3997	7
Lahra, 2012 [43]	Australia	2011	Public and private sector sources	Male and female	Standardised agar plate dilution technique.♦	Standard MIC values for included antimicrobial agents♣	Yes	WHO quality controlled method since 1993.	4230	7
Lahra, 2013 [40]	Australia	2012	Public and private sector sources	Male and female	Standardised agar plate dilution technique.♦	Standard MIC values for included antimicrobial agents♣	Yes	WHO quality controlled method since 1993.	4718	7
Lahra, 2015 [44]	Australia	2013	Public and private sector sources	Male and female	Standardised agar plate dilution technique.♦	Standard MIC values for included antimicrobial agents♣	Yes	WHO quality controlled method since 1993.	4897	7
Lahra, 2015 [31]	Australia	2014	Public and private sector sources	Male and female	Standardised agar plate dilution technique.♦	Standard MIC values for included antimicrobial agents♣	Yes	WHO quality controlled method since 1993.	4808	7
Lahra, 2017 [42]	Australia	2015	Public and private sector sources	Male and female	Standardised agar plate dilution technique.♦	Standard MIC values for included antimicrobial agents♣	Yes	WHO quality controlled method since 1993.	5411	7
Lahra, 2018 [41]	Australia	2016	Public and private sector sources	Male and female	Standardised agar plate dilution technique.♦	Standard MIC values for included antimicrobial agents♣	Yes	WHO quality controlled method since 1993.	6378	7
Lahra, 2019 [28]	Australia	2017	Public and private sector sources	Male and female	Standardised agar plate dilution technique.♦	Standard MIC values for included antimicrobial agents♣	Yes	WHO quality controlled method since 1993.	7835	7
Heffernan, 2004 [38]	New Zealand	April–August 2002	Community and hospital laboratories	Males and females	Standardised agar plate dilution technique.♦	Standard MIC values for included antimicrobial agents♣	Yes	Not stated but used the AGSP method based on WHO quality control method.	615	7
Lee, 2018 [46]	New Zealand	2014-2015	Public and private sector sources	Males and females	Agar dilution method according to Clinical and Laboratory Standards Institute (CLSI ) Gonorrhoea guidelines.	Standard MIC values for included antimicrobial agents♣	Yes	N. gonorrhoea drug-susceptible strain ATCC 49226 and strain NCTC 13479 (WHO K)	398	7
Goire, 2017 [37]	Sydney, Australia	March–June 2015	PublicSector STD clinics.	Male and female	Etest method - MALDI-TOF MS (Bruker Daltonics, Bremen, Germany)	Standard MIC values for included antimicrobial agents♣	Yes	WHO quality controlled method since 1993.	615	7
Brett, 1992 [36]	New Zealand	1988	Public and private sector	Male and female	Standardised agar plate dilution technique.♦	Standard MIC values for included antimicrobial agents♣	Yes	Not indicated	486	6

♦ The standardised agar plate dilution techniques of the Australian Gonococcal Surveillance Programme (AGSP) [22]. ♣ Standardised minimum inhibitory concentration (MIC) values across included studies: For **penicillin**, the following classification was used: —"Susceptible" (MIC), ≤0.03 mg/L " reduced susceptibility", MIC 0.06–0.5 mg/L "resistant", MIC ≥ 1 mg/L (except for AGSP reports prior to 1998 where MIC of 0.004–0.016 mg/L) was susceptible, MIC 0.06–0.25 mg/L reduced susceptibility, and resistant = MIC 1.0 mg/L or more; or else as PPNG strains—PPNG were classified separately. Chromosomally-mediated resistant *Neisseria gonorrhoeae*—CMRNG: "susceptible" MIC< = 0.03 mg/L, "reduced susceptibility" MIC 0.06–0.5 mg/L, "resistant" MIC > = 1 mg/L. **Ciprofloxacin**: reduced susceptibility 0.06–0.5 mg/L, resistant—MIC ≥ 1 mg/L. **Spectinomycin** "susceptible" MIC < 64 mg/L, "resistant" MIC = 128 mg/L. **Ceftriaxone** "susceptible" MIC < 0.03 mg/L. **Cefpodoxime**: "susceptible" MIC <0.001 to 0.25 mg/L. **Azithromycin** "susceptible" MIC ≤ 0.25 mg/L, "reduced susceptibility" = MIC of 0.5 mg/L and "resistant" MIC ≥ 0.5 mg/L. **Tetracycline** "susceptible" MIC = 0.5 mg/L, "resistant" MIC = 1 mg/L, **high level tetracycline resistance** (TRNG), MIC = 32 mg/L. MALDI-TOF MS - Matrix Assisted Laser Desorption/Ionization." Time of Flight Mass Spectrometry (Bruker Daltonics, Bremen, Germany). ATCC 49226 and strain NCTC 13479 (WHO K) - American Type Culture Collection (ATCC) 49226 and strain National Collection of Type Cultures (NCTC) 13479 (WHO NG reference strain K).

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
