# Peer review of "Trends in Antimicrobial Resistance Patterns in Neisseria Gonorrhoeae in Australia and New Zealand: A Meta-analysis and Systematic Review"

_antibiotics, 2019, doi:10.3390/antibiotics8040191_

Round 1

Reviewer 1 Report

This is sound analysis of the trends of antimicrobial resistance across australia over a long period dating from 1981. 

I did not have any major issues with the analysis but it would have helped impact if you had reprented your data in a more accessible manner for people to use the figures. 

Throughout the manuscript there were many small typographical errors which i will not list all here. But they included the use of gonorrhoeae (the species name) incorrectly- the disease is gonorrhoea. There were many instances where internal captial letters were present in sentences and need to be removed. One major point that did not appear in the disucssion is the acknowledgement that many AMR isolates are introduced by international travellers or induced by local treatment protocols. Therefore, the noisiness of the data is really about the contribution of international travel to this picture and this has substantially increased as the costs of flights have reduced over the period of this study. It is most likely the case that the increased proportion of AMR strains over time is due to the increaed frequecy of introductions into the local population rather than local treatment driving resistance. But more detailed surveillance including prescription would be needed to do this. One apsect which could help you there is the acknowlegement that the recommended prescription protocols changed in metropolitan areas over the period of your study.

Author Response

Reviewer 1

This is sound analysis of the trends of antimicrobial resistance across Australia over a long period dating from 1981.

Thank you for your comment.

I did not have any major issues with the analysis but it would have helped impact if you had reprinted your data in a more accessible manner for people to use the figures. We are not quite sure what this Reviewer meant by this comment. However, ew would like to point out that all of the figures embedded in the text as well as the study data tables and additional figures have been made available as supplementary files, which will be accessible to all readers online when the study is published. If this response does not address the reviewer’s concerns, we would appreciate if they could be a bit more specific as to what figures they are referring to and we will make every effort to respond.

Throughout the manuscript there were many small typographical errors which I will not list all here. But they included the use of gonorrhoeae (the species name) incorrectly- the disease is gonorrhoea. There were many instances where internal capital letters were present in sentences and need to be removed.

Thank you. We have picked up the typographical errors which we have now corrected.

One major point that did not appear in the discussion is the acknowledgement that many AMR isolates are introduced by international travellers or induced by local treatment protocols. Therefore, the noisiness of the data is really about the contribution of international travel to this picture and this has substantially increased as the costs of flights have reduced over the period of this study. It is most likely the case that the increased proportion of AMR strains over time is due to the increased frequency of introductions into the local population rather than local treatment driving resistance. But more detailed surveillance including prescription would be needed to do this. One aspect which could help you there is the acknowledgement that the recommended prescription protocols changed in metropolitan areas over the period of your study.

Thanks for the suggestion. We have edited the discussion segment to acknowledge this, by adding comments to the effect of the following: “While the data was not presented in this report, Australian surveillance data indicate that a significant proportion of isolates for which travel history data was available, indicate that infection was either acquired overseas, diagnosed in international travellers or induced by local treatment protocols [17,57]. It is important to also note, that over the more than two and a half decades covered by data in this study, there have been changes to the recommended prescription protocols, particularly in metropolitan areas, in response to resistance patters and in keeping with international guidelines [34,64]. [59] The continued efficacy of combination dual therapy and their effectiveness in delaying the emergence of multi-resistant N. gonorrhoeae has been described [34,59]. It is therefore likely that an important contributing factor to the increase in the proportion of resistant strains over time is the increased frequency of and continued introductions of resistant strains into the local population rather than local treatment protocols that are driving resistance levels. It is reasonable to assume that some of the noise in the data may reflect the contribution of overseas acquired infections to the overall local picture.”

Reviewer 2 Report

Increasing antibiotics resistance is an clinically important topic. However, the world does not consist only of clinical studies and WHO recommendations. The reasons for decreasing susceptibility to antibiotics are complex and include use of antibiotics in agriculture; the tendency of regulatory authorities to approve increasingly smaller segments of clinical indications; additional high burdens that make drug developmenet increasingly more complex and more expensive; prescription patterns by medical doctors; sub-standard medicines sold in many developing countries in parallel to the official network of pharmacies, the general framework of commercial drug development, and many more.

The only lines where these challenges are mentioned in passing are 283-287. There are many superficial assumptions. Just the short duration of a treatment does not automatically mean that the financial incentives to develop new drug treatments are low. This is complex, and if the authors want to be taken seriously, they need to learn more about drug development. To make the paper acceptable, they need to expand the discussion considerably. The meta-analysis may be methodoligcally sound, but background explanation and discussion are superficial and one-dimensional. As it is, the manuscript offers nothing new and nothing interesting. The authors need to expand considerably the background, the discussion and the conclusions.

Author Response

Reviewer 2

Increasing antibiotics resistance is an clinically important topic. However, the world does not consist only of clinical studies and WHO recommendations. The reasons for decreasing susceptibility to antibiotics are complex and include use of antibiotics in agriculture; the tendency of regulatory authorities to approve increasingly smaller segments of clinical indications; additional high burdens that make drug development increasingly more complex and more expensive; prescription patterns by medical doctors; sub-standard medicines sold in many developing countries in parallel to the official network of pharmacies, the general framework of commercial drug development, and many more.

The only lines where these challenges are mentioned in passing are 283-287. There are many superficial assumptions. Just the short duration of a treatment does not automatically mean that the financial incentives to develop new drug treatments are low. This is complex, and if the authors want to be taken seriously, they need to learn more about drug development. To make the paper acceptable, they need to expand the discussion considerably. The meta-analysis may be methodologically sound, but background explanation and discussion are superficial and one-dimensional. As it is, the manuscript offers nothing new and nothing interesting. The authors need to expand considerably the background, the discussion and the conclusions.

We would like to thank the Reviewer for their comments and for acknowledging the soundness of our methodology. We have found the rest of the response to be quite vague and lacking any specific areas of focus for improvement in the manuscript. The manuscript has set out clear aims and established that the scope includes the changes in susceptibility to various antibiotics against N. gonorrhoeae in Australia and New Zealand over time. We do hope that the improvements made to the background and discussion of the manuscript has addressed the concerns of this reviewer.

Reviewer 3 Report

Authors have analyzed the  Trends in antimicrobial resistance patterns in  Neisseria gonorrhoeae in Australia and New Zealand: a meta-analysis and systematic review. The review is very well written, authors have covered all the possible aspects of antimicrobial resistance. I observed no minor or major points which needs attention. I believe review is exhaustive, lucid. Please accept in present form. Thanks

Author Response

The authors would like to thank this reviewer for assessing our manuscript as being  exhaustive, lucid. This feedback is greatly appreciated.

Reviewer 4 Report

In the manuscript submitted to Antibiotics (code 586693) authors review the trends in antimicrobial resistance patterns in Neisseria gonorrhoeae in Australia and New Zealand. This reviewer suggest the publication in Antibiotics after minor revision.

Is the opinion of this reviewer that this review could be useful for university teachers, professionals of the clinical and medical area and its students. Also the number of cites is adequated.

As a minor comment, the Table 1 is tedious and must be rewritted in order to obtain the information in an easily form and time.

Author Response

Thank the Reviewer for their appreciation for the work we have done on this manuscript. We appreciate the comment with regards to the Table. We have shortened the table by condensing the words in Columns 6 and 7. The full explanation for each of these columns is now added as two footnotes at the bottom of the table instead. Footnotes outline below -

¨ The standardised agar plate dilution techniques of the Australian Gonococcal Surveillance Programme [22]

§ Standardised MIC values across included studies: For penicillin, the following classification was used:- "Susceptible" (MIC), ≤0.03 mg/l " reduced susceptibility", MIC 0.06-0.5 mg/l "resistant", MIC ≥ 1 mg/l (except for AGSP reports prior to 1998 where MIC of 0.004-0.016 mg/l) was susceptible, MIC 0.06-0-25 mg/I reduced susceptibility, and resistant = MIC 1.0 mg/l or more; or else as PPNG strains - PPNG were classified separately. CMRNG: "susceptible" MIC<= 0.03 mg/L, "reduced susceptibility" MIC 0.06-0.5mg/L, "resistant" MIC >= 1mg/L. Ciprofloxacin: reduced susceptibility 0.06-0.5 mg/L, resistant - MIC ≥1 mg/L. Spectinomicin "susceptible" MIC <64mg/l, "resistant" MIC =128 mg/L. Ceftriaxone "susceptible" MIC <0.03 MG/L. Cefpodoxime: "susceptible" MIC <0.001 to 0.25 mg/L. Azithromycin "susceptible" MIC ≤0.25 mg/L, "reduced susceptibility" = MIC of 0.5 mg/L and "resistant" MIC ≥ 0.5 mg/L. Tetracycline "susceptible" MIC =0.5 mg/L, "resistant" MIC =1 mg/L, high level tetracycline resistance (TRNG), MIC = 32 mg/L.

Reviewer 5 Report

The manuscript titled “Trends in antimicrobial resistance patterns in Neisseria gonorrhoeae in Australia and New Zealand: a meta-analysis and systematic review” faces an important topic concerning the human health. Focusing attention on Australia and New Zealand, the authors report a wide and detailed collection of data on N. gonorrhoeae infections in order to generate a dataset useful to understand the resistance pattern trends related to N. gonorrhoeae in the space of 37 years.

The knowledge of data is needed to fight the development of antimicrobial resistance.

Overall the manuscript is well-written and aims and scope of the authors is clear. The issue of the antimicrobial resistance is correctly introduced and the experiments have been widely described.

I suggest some minor corrections regarding some inaccuracies:

Row 122: please change “gonorrhea Improved” in “gonorrhea. Improved”; Table 1, line Tapsall, 1996 [50]: change “ciprofloxacin; lless” in “ciprofloxacin; less”; Table 1, line Tapsall 1997 [34]: change “fullysensitive” in “fully sensitive”; Row 180: remove “from” after “between”; Row 266: please write N. gonorrhoeae in italic; Row 311: please change “can reduced” in “can reduce”.

Author Response

We would like to thank the reviewer for their feedback and we appreciate their kind works in regards to the quality of our manuscript. We have corrected the minor issues identified in the attached revised manuscript.  

Round 2

Reviewer 2 Report

There is no significant improvement of this manuscript's version compared to the first manuscript.

Author Response

This reviewer did not provide any items for correction.